# Spatiotemporal Dynamics, Evolutionary History and Zoonotic Potential of Moroccan H9N2 Avian Influenza Viruses from 2016 to 2021

**DOI:** 10.3390/v14030509

**Published:** 2022-03-01

**Authors:** Fatiha El Mellouli, Mohamed Mouahid, Alice Fusaro, Bianca Zecchin, Hasnae Zekhnini, Abderrazak El Khantour, Edoardo Giussani, Elisa Palumbo, Hamid Rguibi Idrissi, Isabella Monne, Abdelaziz Benhoussa

**Affiliations:** 1Biodiversity, Ecology and Genome Laboratory, Faculty of Sciences, Mohammed V University in Rabat, 4 Avenue Ibn Battouta, Rabat 10106, Morocco; hrguibi@hotmail.com (H.R.I.); benhoussa@hotmail.com (A.B.); 2Mouahid’s Veterinary Clinic, Temara 12000, Morocco; mohamedmouahid@gmail.com; 3Istituto Zooprofilattico Sperimentale delle Venezie, 35020 Legnaro, Italy; afusaro@izsvenezie.it (A.F.); bzecchin@izsvenezie.it (B.Z.); egiussani@izsvenezie.it (E.G.); epalumbo@izsvenezie.it (E.P.); imonne@izsvenezie.it (I.M.); 4Immunology and Biodiversity Laboratory, Faculty of Science Ain chock, Hassan II University of Casablanca, Casablanca 20100, Morocco; ha.zekhnini@gmail.com; 5Division of Pharmacy and Veterinary Inputs, Food Safety National Office (ONSSA), Rue Ikhlas Cym, PB 4509 Akkari, Rabat 10050, Morocco; abder_elkhantour@yahoo.fr; 6Ecole supérieure de Technologie de Lâayoune, Quartier 24 Mars, Lâayoune 3007, Morocco

**Keywords:** avian influenza virus, H9N2, phylogeny, molecular evolution, Morocco

## Abstract

The H9N2 virus continues to spread in wild birds and poultry worldwide. At the beginning of 2016, the H9N2 Avian influenza virus (AIV) was detected in Morocco for the first time; despite the implementation of vaccination strategies to control the disease, the virus has become endemic in poultry in the country. The present study was carried out to investigate the origins, zoonotic potential, as well as the impact of vaccination on the molecular evolution of Moroccan H9N2 viruses. Twenty-eight (28) H9N2 viruses collected from 2016 to 2021 in Moroccan poultry flocks were isolated and their whole genomes sequenced. Phylogenetic and evolutionary analyses showed that Moroccan H9N2 viruses belong to the G1-like lineage and are closely related to viruses isolated in Africa and the Middle East. A high similarity among all the 2016–2017 hemagglutinin sequences was observed, while the viruses identified in 2018–2019 and 2020–2021 were separated from their 2016–2017 ancestors by long branches. Mutations in the HA protein associated with antigenic drift and increased zoonotic potential were also found. The Bayesian phylogeographic analyses revealed the Middle East as being the region where the Moroccan H9N2 virus may have originated, before spreading to the other African countries. Our study is the first comprehensive analysis of the evolutionary history of the H9N2 viruses in the country, highlighting their zoonotic potential and pointing out the importance of implementing effective monitoring systems.

## 1. Introduction

The H9N2 AIV was initially isolated from turkeys in Wisconsin in the United States of America in 1966 (A/turkey/Wisconsin/1/1966(H9N2)) [1]. In Asia, the first cases of H9N2 infection in poultry were recorded in Hong Kong in the 1970s [2]. In the 1990s, the H9N2 spread became more extensive with several outbreaks affecting industrial poultry in China, Germany, South Korea, Italy, Iran, Pakistan, the USA, South Africa, and Ireland [3,4,5]. This lead to a persistent viral circulation in many countries in Asia, the Middle East, and North Africa [6], where the virus has attained endemic status in poultry populations [7,8,9]. Today, H9N2 is considered the most prevalent, and damaging, low-pathogenic avian influenza (LPAI) subtype in poultry worldwide [7], presenting a threat to both animal and human health [9,10]. H9N2 infections usually cause minor clinical disease in poultry, which makes its identification in the field rather challenging [11]. However, despite their low pathogenicity, H9N2 viruses may cause variable pathogenicity and mortality in domestic poultry [5,6,12], especially when associated with immunosuppression or coinfection with other viruses or bacteria [3,11,13,14], resulting in significant economic losses [5,6,12].

The H9N2 virus has a zoonotic potential with interspecies transmission between poultry and mammals, including swine and humans [15,16,17,18]. At least 59 human cases have been reported to date [9]. Furthermore, H9N2 viruses have proved able to provide partial or even complete sets of internal gene segments to zoonotic avian influenza viruses (AIVs), leading to the emergence of human-lethal reassortants, as in the case of China in 2013 with the appearance of H7N9 and H10N8 viruses, and, a year later of the H5N6 subtype [19,20,21,22,23,24,25,26] posing serious risks to public health. Moreover, there are multiple reports of H9N2 natural infection in other mammals, such as ferrets [27], mice [28], minks, foxes, raccoon dogs [29], cats, and dogs [30].

Based on their genetic characteristics, the H9N2 AIVs are divided into three major lineages: the G1-like lineage (A/Quail/Hong Kong/G1/97), the Korean-like lineage (A/Chicken/Korea/38349-p96323/96 and A/Duck/Hong Kong/Y439/97) and the Y280-like lineage (A/Duck/Hong Kong/Y280/97, A/Chicken/Beijing/1/94, and A/Chicken/Hong Kong /G9/97) [3,4,5,31]. Until now, the G1-like lineage has been the only one found in Africa [32].

Since the early 2000s, the emergence of H9N2 viruses has been reported in several North African countries, including Libya in 2006 [33], Egypt in 2006 [34], and Tunisia in 2009 [35]. Additionally, following the first isolation of the LPAI H9N2 virus in Morocco in January 2016 [36], H9N2 viruses have been extensively circulating in North Africa, spreading to West Africa, as well as to East Africa. Since 2017, H9N2 viruses have been detected in Algeria [37] and in several Sub-Saharan African countries, including Burkina Faso [38], Ghana [32], Uganda, Kenya, Senegal [39], Benin Republic, and Togo [40]. All H9N2 viruses detected in poultry in Africa belong to the G1 lineage and are related to those isolated in the United Arab Emirates.

In Morocco, following its first identification in 2016, the H9N2 virus has spread rapidly across the country, causing significant economic losses in poultry [36]. A previous study revealed that the Moroccan isolate might be related to the Middle East isolate A/chicken/Dubai/D2506.A/2015 [41]. The virus showed high pathogenicity under field conditions with aggravation of the disease in case of coinfection with other respiratory diseases [41,42]. Despite the swift implementation of a mass poultry vaccination program following the onset of the disease, the virus became endemic in all poultry sectors of the country [43]. Knowledge of the evolutionary history and the circulation of the Moroccan H9N2 viruses is crucial for public and animal health in Morocco. A better understanding of the epidemiological and geographic origins of the Moroccan H9N2 viruses might help to improve the control and management of future outbreaks. In this study, we characterized the whole genome of H9N2 AIVs collected in Morocco from 2016 to 2021, exploring the spatiotemporal dynamics and the acquisition of mutations of particular concern for animal and human health.

## 2. Materials and Methods

### 2.1. Sample Collection

From 2016 to 2021, thanks to the establishment of a program to monitor the presence of influenza viruses in poultry in Morocco, samples were collected from 81 poultry farms located in seven different regions in the country, including layers, breeders, turkeys, and broiler flocks of varying ages. Most flocks (72.84%, 59/81) were vaccinated against AIV H9N2. Oropharyngeal, cloacal swabs and organ samples were collected from chickens and turkeys, stored at 2–8 °C, and sent to the laboratory within 24 h for further analysis.

### 2.2. Real Time RT-PCR for H9 Gene Detection

Swabs, as well as tissue samples, were re-suspended in 1 mL of phosphate-buffered saline (pH 7.4 +/− 0.2, supplemented with 10,000 IU/mL penicillin, 10 mg/mL streptomycin, 0.25 mg/mL gentamycin, and 5000 IU/mL nystatin), thoroughly homogenized by using Bead Blaster 24 Homogenizer (Benchmark Scientific, 2600, Sayreville, NJ, USA). After centrifugation at 1500× *g* for 10 min at 4 °C, total RNA was purified with the NucleoSpin^®^ RNA Virus Kit (Macherey-Nagel, Düren, Germany) following the manufacturer’s instructions. RNA was then eluted in 50 µL RNase-free water. All samples were tested by RT PCR for the influenza A matrix gene as described by Spackman, E. et al. (2002) [44]. The positive samples were then tested for the H9 gene detection by RT-PCR according to Monne, I. et al. (2008) [45] by using the AgPath-IDTM One-Step RT-PCR Kit (Ambion, Applied Biosystems, Grand Island, NY, USA). The rRT-PCR was performed with Aria Mx (Agilent Technologies, Santa Clara, CA, USA) and ABI 7500 Fast (Applied Biosystems, Foster City, CA, USA) 96-well plate real-time PCR devices.

### 2.3. Whole Genome Sequencing

The complete genomes were amplified using SuperScript™ III One-Step RT-PCR System with Platinum™ Taq High Fidelity DNA Polymerase (Invitrogen, Carlsbard, CA, USA), as previously described by Zhou B. et al. (2009) [46]. Amplicons were purified by using Agencourt AMPure XP (Beckman Coulter Inc., Brea, CA, USA) and quantified with Qubit™ DNA HS Assay (Thermo Fisher Scientific, Waltham, MA, USA). Sequencing libraries were prepared with the Illumina Nextera XT DNA Sample Preparation Kit (Illumina, San Diego, CA, USA) and sequenced on the Illumina MiSeq platform (2 × 250 bp Paired-End; Illumina, San Diego, CA, USA).

Read quality was assessed using FastQC v0.11.2 (https://www.bioinformatics.babraham.ac.uk/projects/fastqc/, accessed on 4 June 2021) and raw data were filtered by removing reads with more than 10% of undetermined bases, reads with more than 100 bases with Q score below seven, and duplicated paired-end reads. Remaining reads were clipped from Illumina Nextera XT adaptors with scythe v0.991 (https://github.com/vsbuffalo/scythe, accessed on 4 June 2021) and trimmed with sickle v1.33. (https://github.com/najoshi/sickle, accessed on 4 June 2021). Reads were cropped to a maximum of 250 bases length using trimmomatic [47]. Reads shorter than 80 bases or unpaired after previous filters were discarded. High-quality reads were aligned against a reference genome using BWA v0.7.12 [48]. Alignment was processed using Picard-tools v2.1.0 and GATK v3.5 [49,50] in order to correct potential errors, realign reads around indels and recalibrate base quality. LoFreq v2.1.2 [51] was used to call Single Nucleotide Polymorphisms (SNPs). Consensus sequences were submitted to the GISAID EpiFlu™ database (http://www.gisaid.org, accessed on 6 January 2022) under the accession numbers listed in Table 1.

### 2.4. Phylogenetic and Evolutionary Analyses

After sequence alignment using MAFFT v7 [52], IQTREE v1.6 was used to perform ultrafast bootstrap resampling analyses (1000 replications) to generate the maximum likelihood phylogenetic trees [53,54]. Trees were visualized with FigTree v1.4.4 (http://tree.bio.ed.ac.uk/software/figtree/, accessed on 30 June 2021). 

Phylogenetic network analyses using the Median Joining method implemented in the program NETWORK 10 (https://www.fluxus-engineering.com/, accessed on 4 June 2021) were performed on the HA gene sequences of the H9N2 influenza viruses collected in Morocco in the period 2016–2021 [55]. This method uses a parsimony approach to reconstruct the relationships between highly-similar sequences, and allows the creation of “median vectors,” which represent un-sampled sequences used to connect the existing genotypes in the most parsimonious way.

The evolutionary rate and the time to the most common recent ancestor (tMRCA) were estimated for all the gene segments using BEAST v1.10.4 [56]. Markov chain Monte Carlo (MCMC) sampling analyses employing an uncorrelated lognormal relaxed molecular clock, that allows for rate variation across lineages, and the SRD06 substitution model (HKY85 + Γ4 with two partitions—1st + 2nd positions vs. 3rd position -, base frequencies, and Γ-rate heterogeneity unlinked across all codon positions) were implemented [57]. Tree Annotator v1.10.4 (http://beast.bio.ed.ac.uk/TreeAnnotator/, accessed on 11 August 2021) was used to summarize the Maximum Clade Credibility (MCC) trees, which were then visualized in FigTree v1.4.4 (http://tree.bio.ed.ac.uk/software/figtree/, accessed on 11 August 2021).

### 2.5. Analysis of Selection Pressures

Site-specific selection pressures for the HA gene segment of the Moroccan H9N2 viruses were measured as the ratio of nonsynonymous (dN) to synonymous (dS) nucleotide substitutions per site. Specifically, the Mixed Effects Model of Evolution (MEME) and the fixed-effects likelihood (FEL) methods were used to infer sites subjected to episodic or pervasive positive selection [58,59]. Analyses were performed using the Datamonkey online version of the Hy-Phy package [60]. All analyses used the GTR model of nucleotide substitution and ML phylogenetic trees.

## 3. Results

### 3.1. Phylogenetic and Evolutionary Analyses

In the framework of the surveillance activity implemented in Morocco between 2016 and 2021, field samples collected from 81 poultry farms were submitted to the laboratory to be tested for the presence of AIV. Overall, 76 out of 81 (76/81) tested positive for AIV and specifically for the H9 subtype (Figure 1 shows the location of the sampled and the positive farms). In total, 28 representative positive samples were selected based on the year and area of collection for whole genome sequencing. Specifically, we obtained the complete genome of 22 viruses and the partial genome of 6 viruses (Table 1).

Phylogenetic analysis of the hemagglutinin (HA) gene segment revealed that all the H9N2 viruses collected in the country since 2016 belong to the G1 lineage and cluster with H9N2 viruses identified in the Middle East (United Arab Emirates and Oman), North Africa (Algeria), and West Africa (Burkina Faso, Senegal, Togo, Ghana, Benin) between 2015 and 2020 (Figure 2). The phylogenetic trees of the other gene segments confirm the same genetic clustering (see Appendix A). The tree topologies indicate that the H9N2 viruses from Morocco had evolved into multiple genetic groups within the country. However, the lack of a consistency of the groups among the different phylogenies is suggestive of the occurrence of multiple inter-subtype reassortment events.

Virus evolution into multiple genetic clusters is evident also from the HA median-joining network analysis. Although all the 2016–2017 sequences (yellow in Figure 3) show a high genetic relationship (from 1 to 16 nucleotide differences) and cluster together on the centre of the network, the viruses collected in 2018–2019 (green in Figure 3) and 2020–2021 (blue in Figure 3) separate from the 2016–2017 progenitor viruses by long branches (from 15 to 54 nucleotide differences) that spread from the centre in multiple directions, forming separate clusters. The high number of median vectors, representing the lost ancestral sequences, clearly indicate a relevant data gap. No particular grouping was observed according to the geographical origin, which suggests a wide virus dispersal within the country.

To investigate the role of Morocco in the geographic spread of the H9N2 viruses of the G1 lineage, we performed a Bayesian phylogeographic analysis. Our results showed a virus spread from the Middle East to Morocco, and subsequently from Morocco to Algeria and from Morocco to West Africa. All of these transitions were well supported by Bayes Factors higher than 20 (BF > 20, Figure 4). This finding suggests that Morocco might have played a central role in the spread of the H9N2 virus in the African continent, although this interpretation may have been biased by the higher number of sequences from Morocco compared to the ones from other African countries.

### 3.2. Molecular Analysis and Positively Selected Sites

The molecular analysis revealed that the H9N2 viruses from Morocco present the same cleavage site (RSSR/GLF) as the previously identified North African H9N2 viruses of the G1 lineage. In addition, most of the Moroccan viruses possess the amino acid Leucine at position 226 and Threonine at position 155 (H3 numbering) of the HA receptor binding site; these mutations imply preferential binding to human-like α2-6-linked sialic acid (SA α2–6) receptors [61,62]. Of note, site 226 has been identified as under positive selection by both MEME and FEL methods (*p*-value < 0.05) (Table 2). The H9N2 Moroccan viruses show three different amino acids in this position: Leucine (L), Glutamine (Q), and Arginine (R). These amino acid substitutions may affect not only the virus receptor binding properties but also influence its antigenic specificity [63]. Although amino acids L and Q have frequently been identified in viruses belonging to the G1 lineage, R has rarely been observed in the H9N2 viruses (one virus from Algeria and one from Morocco). Previous studies demonstrated that this mutation is associated with change of the pandemic 2009 (H1N1) viral HA binding preference, from the human-type receptor, α2,6-linked sialic acid, to the avian-type receptor, α2,3-linked sialic acid [64,65].

In addition to 226, in the HA gene we identified three further sites (sites 190, 160, and 186) under positive selection: one (site 190) using both methods, while the other two were recognized only by FEL. Of note, all these sites fall within the receptor-binding domain and/or antigenic sites [63,66,67].

## 4. Discussion

In Morocco, the first outbreak of LPAI H9N2 was described in broiler flocks in January 2016 [36]. The virus was later isolated from layers and breeders in different regions of the country. As part of a surveillance program, the national veterinary services implemented mass vaccination against H9N2 using homologous inactivated vaccines in all sectors of poultry production, first for breeders and layers, and a few days later for turkey and broiler flocks. The outbreak was controlled in 3–4 months, after which the vaccination was authorized for all types of poultry production using autogenous and heterologous inactivated vaccines. Over the past 5 years, 13 H9N2 vaccines originating from different H9 strains, i.e., from Morocco, UAE, Korea, Turkey, Italy, Egypt, were available and commercialized in Morocco (see Appendix A). Despite this vaccine availability, a recent study on the effectiveness of the four most used vaccines in Moroccan poultry farms has confirmed the need to continuously adapt the vaccine strains to the virus circulating in the field [69]. In the neighboring North African countries, vaccination against AIV H9N2 is also implemented in poultry production sectors using different vaccine types, such as inactivated homologous vaccines and vectored vaccines in Algeria. In Tunisia, vaccination using inactivated homologous vaccines for chicken and turkey breeders and layers has been performed since 2015, but to the present day, no vaccination has been used in broiler flocks. In Egypt, vaccination is currently implemented using local and non-local strains, with variable efficacy. In contrast, vaccination in the field has never been reported in Libya [70]. Despite such vaccination approaches, LPAI H9N2 has become enzootic in North Africa and continues to cause economic losses even in the vaccinated flocks. Thus, circulating H9N2 viruses in North Africa need to be further investigated, and at the same time, revision of employed vaccines and of regional surveillance activities should be implemented.

Our results indicate that all Moroccan H9N2 viruses from 2016 to 2021 belong to the G1 like-lineage and are related to the viruses from the Middle East and North and West Africa, as described in previous reports [43,71]. Results obtained from the spatial analysis demonstrated that AIV H9N2 had been introduced into Morocco from the Middle East, probably through the import of falcons for hunting, and has been circulating in the country for the last six years. The study also showed that the H9N2 G1 like-lineage is the predominant H9 genotype in Moroccan poultry farms. From Morocco, the virus spread to Algeria and West Africa, suggesting the key role of Morocco in the H9N2 virus dispersal on the continent, as reported in previous studies [32,37,38,39,40]. Such virus-spread might have occurred through the movement of poultry professionals [38] and/or the trade of poultry products, since the export market of day-old-chicks and hatching eggs from Morocco to West Africa is well documented [72]. However, the gap in epidemiological data, as well as the limited number of sequences available from the other African countries, might have affected our results and biased our interpretation. Evolutionary and network analyses of the H9N2 viruses from Morocco indicate a high similarity among all the 2016–2017 HA sequences, which is in accordance with previous studies [43,71]. Long branches separate the 2018–2019 and 2020–2021 viruses from their ancestors (2016–2017). This might be explained by the implementation of vaccination from 2016 onwards. The high vaccination coverage and the use of multiple types of vaccines may have driven the virus evolution in multiple directions, with the consequent generation of multiple separate genetic groups in the country. This was further confirmed by the identification of several sites in the HA protein under positive selection located in the RBS or antigenic sites, which may facilitate vaccine escapes, considering the fact that most (71.05%, 54/76) of the H9N2 positive flocks were vaccinated.

As previously observed in the H9N2 African viruses [32,38,40,73], most of Moroccan H9N2 HA protein harbored mammalian-adaptive mutations, including the hemagglutinin human-like marker 216L (226 in H3 numbering), which promotes preferential binding to the α2–6-linked sialic acid receptors of human epithelial cells [60] and was described as an immune escape mechanism [62]. The evidence for positive selection of the 226L marker during serial passage of the virus in chickens was provided by Jegede et al., 2018 [74], suggesting that the H9N2 viruses acquired the mammalian adaptation while circulating in poultry. However, specific receptor-binding studies will be instrumental to assess the zoonotic potential of the H9N2 Moroccan viruses, as other mutations in the HA and internal proteins may be necessary for a successful avian to human adaptation. The present work is the first comprehensive study that provides an overview of the historical origins and evolutionary features of Moroccan H9N2 AIV, based on the Bayesian phylodynamic analysis and whole genome characterization. Over the past five years, the H9N2 AIV in Morocco has shown a rapid genetic diversification, likely directed by selective pressure exerted by vaccine-induced antibodies. The observed mutations and reassortments might extend the range of host species and could lead to the emergence of new strains [75,76]. In addition, our study demonstrates that viruses with a marker well-described in literature associated with zoonotic potential (226L, H3 numbering) may represent a serious risk for public health in Morocco. Hence, the circulation of H9N2 AIV should be investigated in other species, such as wild birds and mammals, including humans, to better understand virus ecology in Morocco. All the more so as a recent molecular study on the presence of the AIV in wild birds staying in the corridor of wetlands during their winter crossing in Morocco, showed a high positivity-rate of the influenza virus in different birds species [77]. Continuous systematic surveillance of molecular evolution of H9N2 AIV is strongly needed to anticipate the emergence of novel H9N2 strains escaping vaccine protection or with an increased human pandemic potential.

## Figures and Tables

**Figure 1 viruses-14-00509-f001:**
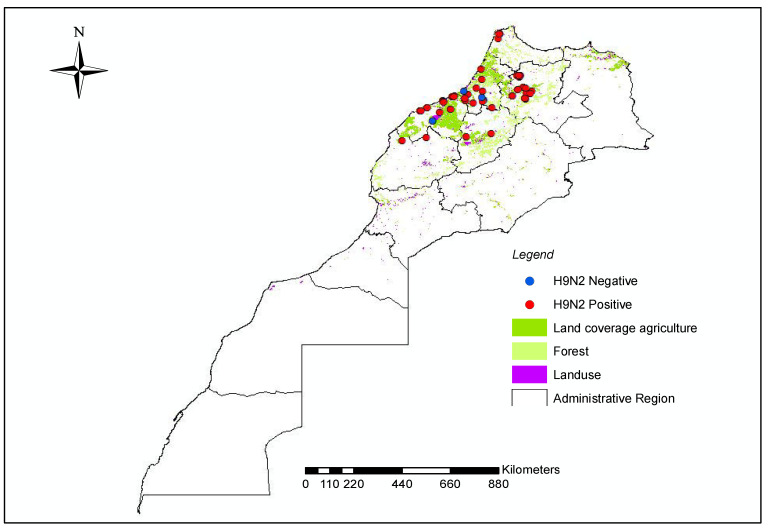
Location of the sampled and positive farms.

**Figure 2 viruses-14-00509-f002:**
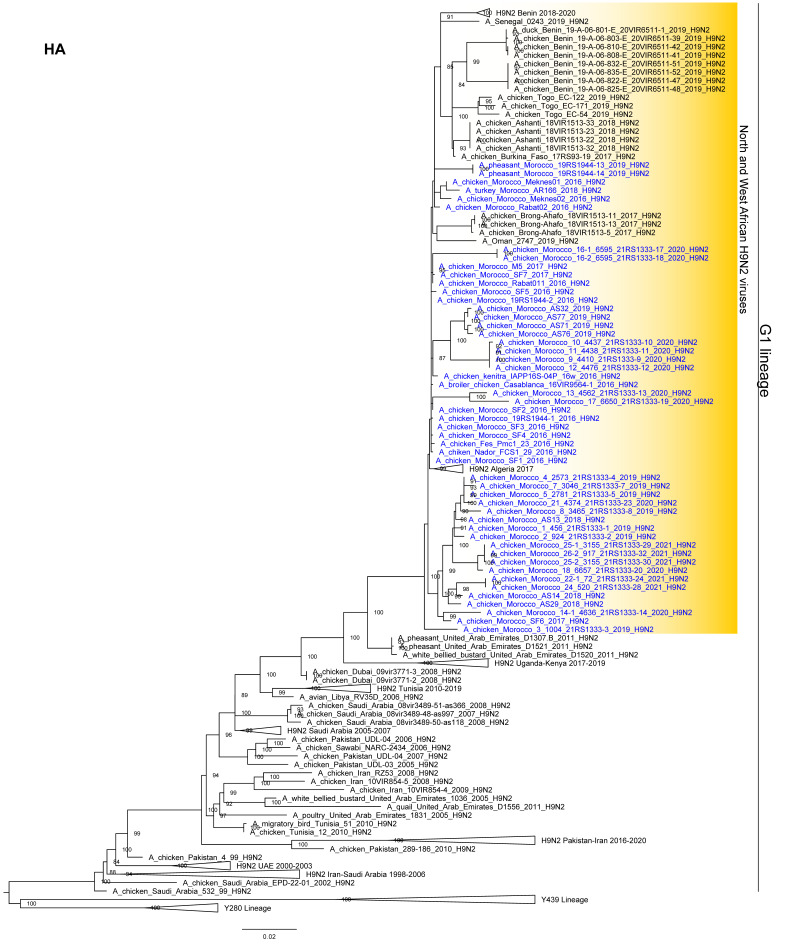
Maximum Likelihood phylogenetic tree of the HA gene (IQ-TREE v.1.6.8). The H9N2 viruses from Morocco analyzed in this study are marked in blue.Ultra-fast bootstrap supports equal to or higher than 80% are indicated next to the nodes.

**Figure 3 viruses-14-00509-f003:**
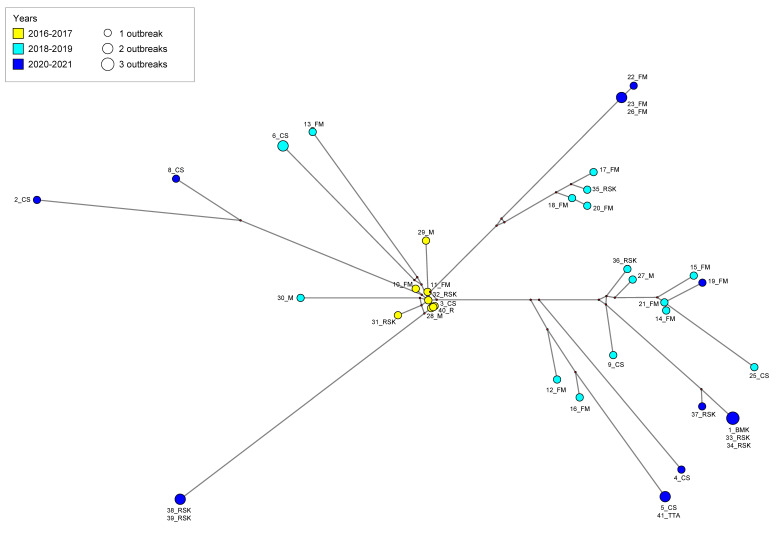
The HA median-joining network showing high genetic relationship between all 2016 and 2017 H9N2 Moroccan viruses while the viruses of 2018–2019 and 2020–2021 are separated from their 2016–2017 ancestors by long branches.

**Figure 4 viruses-14-00509-f004:**
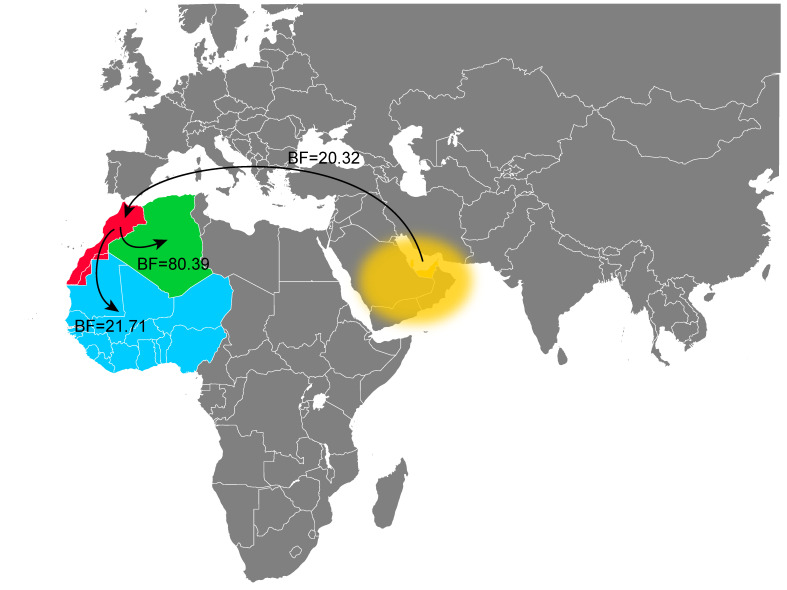
Spatiotemporal dynamics describing the geographic origins of Moroccan H9N2 viruses and the virus spread from Morocco to other African countries. Supported transitions are indicated by continuous lines (BF > 20).

**Table 1 viruses-14-00509-t001:** H9N2 isolates related information and GISAID accession numbers.

Year	Virus	Subtype	Flock	Age (d/w ^1^)	Collection Date	Region	Location	GISAID Accession Number
2016	A/chicken/Morocco/19RS1944-1/2016	H9N2	Breeders	na	25 January 2016	Fès-Meknès	Fès	EPI1950238–EPI1950245
	A/chicken/Morocco/19RS1944-2/2016	H9N2	Broilers	na	26 January 2016	Rabat-Salé-Kénitra	Kenitra	EPI1950325–EPI1950332
2019	A/pheasant/Morocco/19RS1944-13/2019	H9N2	na	na	19 March 2019	Casablanca-Settat	Ain Borja	EPI1950209–212, 214-216 and EPI1950384
	A/pheasant/Morocco/19RS1944-14/2019	H9N2	na	na	19 March 2019	Casablanca-Settat	Ain Borja	EPI1950217–EPI1950224
	A/chicken/Morocco/1_456_21RS1333-1/2019	H9N2	Layers	26 w	16 February 2019	Rabat-Salé-Kénitra	Skhirat Temara	EPI1950349–EPI1950356
	A/chicken/Morocco/2_924_21RS1333-2/2019	H9N2	Breeders	33 w	3 April 2019	Casablanca-Settat	El Jadida	EPI1950230–233, 235–237 and EPI1950385
	A/chicken/Morocco/3_1004_21RS1333-3/2019	H9N2	Broilers	15 d	11 April 2019	Fès-Meknès	Fès	EPI1950246–EPI1950253
	A/chicken/Morocco/4_2573_21RS1333-4/2019	H9N2	Broilers	34 d	5 September 2019	Fès-Meknès	Sefrou	EPI1950277–EPI1950284
	A/chicken/Morocco/5_2781_21RS1333-5/2019	H9N2	Breeders	29 w	24 September 2019	Fès-Meknès	Fès	EPI1950254–EPI1950261
	A/chicken/Morocco/7_3046_21RS1333-7/2019	H9N2	Broilers	24 d	19 October 2019	Fès-Meknès	Ifrane	EPI1950262–265, 267–269 and EPI1950386
	A/chicken/Morocco/8_3465_21RS1333-8/2019	H9N2	Layers	22 w	29 November 2019	Rabat-Salé-Kénitra	Bouznika	EPI1950309–EPI1950316
2020	A/chicken/Morocco/21_4374_21RS1333-23/2020	H9N2	Breeders	42 w	25 February 2020	Fès-Meknès	Meknès	EPI1950270–EPI1950276
	A/chicken/Morocco/9_4410_21RS1333-9/2020	H9N2	Breeders	43 w	26 February 2020	Fès-Meknès	EL Hajeb	EPI1950317–320, 322–324 and EPI1950387
	A/chicken/Morocco/10_4437_21RS1333-10/2020	H9N2	Broilers	34 d	29 February 2020	Fès-Meknès	Sefrou	EPI1950285–EPI1950292
	A/chicken/Morocco/11_4438_21RS1333-11/2020	H9N2	Broilers	34 d	29 February 2020	Fès-Meknès	Sefrou	EPI1950293–EPI1950300
	A/chicken/Morocco/12_4476_21RS1333-12/2020	H9N2	Layers	33 w	3 March 2020	Rabat-Salé-Kénitra	Bouznika	EPI1950301–EPI1950308
	A/chicken/Morocco/13_4562_21RS1333-13/2020	H9N2	Broilers	34 d	12 March 2020	Casablanca-Settat	El Jadida	EPI1950225–EPI1950229
	A/chicken/Morocco/14-1_4636_21RS1333-14/2020	H9N2	Broilers	27 d	20 March 2020	Casablanca-Settat	Casablanca	EPI1950194–EPI1950200
	A/chicken/Morocco/16-1_6595_21RS1333-17/2020	H9N2	Broilers	39 d	23 November 2020	Rabat-Salé-Kénitra	Rommani	EPI1950365–EPI1950369
	A/chicken/Morocco/16-2_6595_21RS1333-18/2020	H9N2	Broilers	39 d	23 November 2020	Rabat-Salé-Kénitra	Rommani	EPI1950370–EPI1950375
	A/chicken/Morocco/17_6650_21RS1333-19/2020	H9N2	Broilers	41 d	28 November 2020	Rabat-Salé-Kénitra	Bouznika	EPI1950186–EPI1950193
	A/chicken/Morocco/18_6657_21RS1333-20/2020	H9N2	Broilers	37 d	30 November 2020	Rabat-Salé-Kénitra	Skhirat Temara	EPI1950357–EPI1950364
	A/chicken/Morocco/19_6700_21RS1333-21/2020	H9N2	Layers	44 w	4 December 2020	Casablanca-Settat	Casablanca	EPI1950870–EPI1950875
2021	A/chicken/Morocco/22-1_72_21RS1333-24/2021	H9N2	Broilers	21 d	11 February 2021	Tanger-Tétouan- Al Hoceïma	Tangeir	EPI1950376–EPI1950383
	A/chicken/Morocco/24_520_21RS1333-28/2021	H9N2	Broilers	40 d	18 March 2021	Casablanca-Settat	Casablanca	EPI1950201–EPI1950208
	A/chicken/Morocco/25-1_3155_21RS1333-29/2021	H9N2	Broilers	40 d	9 April 2021	Rabat-Salé-Kénitra	Sidi Boubker El Haj	EPI1950333–EPI1950340
	A/chicken/Morocco/25-2_3155_21RS1333-30/2021	H9N2	Broilers	40 d	9 April 2021	Rabat-Salé-Kénitra	Sidi Boubker El Haj	EPI1950341–EPI1950348
	A/chicken/Morocco/26-2_917_21RS1333-32/2021	H9N2	Layers	19 w	28 April 2021	Béni Mellal-Khénifra	Béni Mellal	EPI1950178–EPI1950185

^1^ d: days; w: weeks; na: not available.

**Table 2 viruses-14-00509-t002:** Sites under positive selection (*p*-value < 0.05) in the Moroccan H9N2 viruses HA protein.

Method	Site (H9 Numbering)	Site (H3 Numbering)	*p*-Value	Aminoacids		References
MEME	180	190	0.03	A, T, V	RBS, antigenic site B	[67]
	216	226	0.03	L, Q, R	RBS, antigenic site D	[68]
FEL	180	190	0.02	A, T, V	RBS, antigenic site B	[67]
	216	226	0.02	L, Q, R	RBS, antigenic site D	[68]
	150	160	0.03	G, N, V	antigenic site B	[64]
	176	186	0.04	P, A	RBS	[69]

* H9 numbering.

## Data Availability

The primary data used to support the findings of this study are available from the corresponding author upon request.

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
