# Peer review of "Spatiotemporal Dynamics, Evolutionary History and Zoonotic Potential of Moroccan H9N2 Avian Influenza Viruses from 2016 to 2021"

_viruses, 2022, doi:10.3390/v14030509_

Round 1

Reviewer 1 Report

The work by Mellouli et al entitled "Spatiotemporal dynamics, evolutionary history and zoonotic potential of Moroccan H9N2 Avian Influenza viruses from 2016 to 2021” presents a thorough examination of the genetic makeup of representative H9N2 AIV strains circulating in Morocco. The manuscript is well-written, clear, and easy to understand. Included below are a few minor points for the authors to review.

The authors might consider adding more information about the content of the vaccine (based on the G1 lineage?) to the introduction, in addition to the information presented in the discussion. Also, it would be useful to know the vaccination status of poultry in the surrounding countries.

In the discussion, lines 291-293 contain broad and unspecified information. What mutations were found that increase virulence or are associated with a shift in host specificity? Did the authors examine them in this current work? If so, please discuss in more detail. Without that additional information, the claim in the next paragraph “the present work is the first comprehensive study… of zoonotic potential” seems a bit exaggerated. The authors characterized a well-known RBS specificity residue and confirmed that the AAs reflect the “human-type” orientation. However, without receptor binding studies, conclusions about the actual binding specificity are premature. Additionally, other changes in HA impacting cleavage, fusion stability etc as well as changes in other genes are important for successful avian to mammalian adaptation. Did the authors examine these other sites? Without that data, the authors should consider re-phrasing the zoonosis potential claims to more accurately reflect the analyses presented here.

Author Response

Dear Reviewer,

Thank you for your valuable comments.

Please find attached our response to your comments.

We remain at your entire disposal if there are any other modifications, remarks and suggestions.,

Best regards

Reviewer 2 Report

"Spatiotemporal dynamics, evolutionary history and zoonotic potential of Moroccan H9N2 Avian Influenza viruses from 2016 to 2021" is a classic study of the evolution and geographical distribution of H9N2 avian influenza viruses in Morocco. The work done at a high modern level, the results are beyond doubt. It was shown that the first two years after the introduction of H9N2 in Morocco (2016-2017), the virus evolved slowly, but then, probably due to the wide coverage of poultry vaccination, several lines emerged that rapidly accumulated amino acid substitutions. The substitutions affected antigenic sites and the receptor-binding site of hemagglutinin. In particular, there were substitutions that promote the transition of the virus to mammals. It is particularly interesting that these substitutions occurred during the circulation of the virus in chickens. The work provokes a critical analysis of the current practice of vaccination against low pathogenic avian influenza viruses.

Author Response

Dear Reviewer,

Thank you for your valuable comments.

We remain at your entire disposal if there are any remarks and suggestions.

Best regards
